# Non-Clinical Factors Determining the Prescription of Antibiotics by Veterinarians: A Systematic Review

**DOI:** 10.3390/antibiotics10020133

**Published:** 2021-01-30

**Authors:** Miguel Servia-Dopazo, Margarita Taracido-Trunk, Adolfo Figueiras

**Affiliations:** 1Department of Preventive Medicine and Public Health, Salnés Clinical Hospital, 36619 Vilagarcía de Arousa, Spain; 2Department of Preventive Medicine and Public Health, University of Santiago de Compostela, 15706 Santiago de Compostela, Spain; margarita.taracido@usc.es (M.T.-T.); adolfo.figueiras@usc.es (A.F.); 3Consortium for Biomedical Research in Epidemiology & Public Health (CIBER en Epidemiología y Salud Pública—CIBERESP), 15706 Santiago de Compostela, Spain; 4Institute of Health Research of Santiago de Compostela (IDIS), 15706 Santiago de Compostela, Spain

**Keywords:** veterinarians, antibiotic resistance, antibiotic prescription, one health, animal health, infection control, antibiotic stewardship

## Abstract

The misuse of antibiotics in humans, animals, and plants is related to the spread of resistant antibiotic strains among humans and animals. In this paper, we carry out a bibliographic search of Medline, Web of Knowledge, and Cab Abstracts with the main objective of ascertaining the available evidence on non-clinical factors and attitudes that could influence the prescription of antibiotics by veterinarians. A total of 34 studies fulfilled the inclusion criteria. Whereas, veterinary health professionals’ prescribing habits did not appear to be influenced by their socio-demographic characteristics, they were influenced by different attitudes, such as fear (identified in 19 out of 34 studies), self-confidence (19/34), business factors (19/34), and by complacency (16/34). Certain owner-related factors, such as lack of awareness (16/34) and demand for antibiotics (12/34), were also important, as were concurrent factors, ranging from a lack of appropriate regulations (10/34) to the expense and delays involved in performing culture and sensitivity tests (10/34) and inadequate farm hygiene (8/34). Our results appear to indicate that the non-clinical factors are potentially modifiable. This may be useful for designing interventions targeted at improving antibiotic use in animals, as part of an overall strategy to reduce the global spread of multi-resistant strains.

## 1. Introduction

Antimicrobials are a key resource, and antimicrobial resistance is acknowledged as one of the most serious threats currently facing public health worldwide. The problem of resistance does not only affect human medicine. In addition, while the transfer of resistance between animals and humans continues to be the subject of research, many studies published in recent years nonetheless show that antibiotic (ABX) use in animals can contribute to the appearance of resistant pathogens in animals and humans alike [1,2,3,4,5,6,7].

The amount of antibiotics consumed by animals worldwide is huge, according to some estimates and if measures are not taken, this use will continue to rise around the world, with estimates showing that by 2030 it will be 11.5% higher than in 2017 [8]. Although, it is important to highlight the differences between low and middle-income countries [9] and North America and Europe, where, for the 25 countries reporting sales data to the ESVAC for each year from 2011 to 2017, an overall decrease of 32.5% in sales (mg/PCU) was observed [10]. On the other hand, the use of antimicrobial preparations designed for groups of animals (pre-mixes, oral powders, and oral solutions) continues to be a common practice [10,11]. This mass treatment of animals increases the likelihood of sick and healthy animals being exposed -repeatedly in many instances- to subtherapeutic doses of broad-spectrum ABXs [11], which could further increase the risk of generating resistance [12,13,14,15].

Even though the most recent research concludes that the use of antibiotics in animals contributes to the human antimicrobial burden by only a small degree [16], a recent meta-analysis concludes that interventions which restrict antibiotic use in animals may reduce the prevalence of resistant bacteria in humans [5]. Based on this fact, the latest World Health Organization (WHO) guidelines firmly recommend a general reduction in the use of critically important antimicrobials for human health in animal production [17]. In parallel to this, the OIE (World Organization for Animal Health) has also developed a list of antimicrobials which are of critical importance in veterinary medicine, the use of which should be restricted [18]. It is important to reduce antibiotic use in all sectors and to ensure that antibiotics remain effective as long as possible The WHO and European Union (EU) urge Member States to develop and implement strategies to contain the development of antibiotic resistance from a joint human, plant, and veterinary perspective, the so-called “One Health approach”, a collaborative, multisectoral, and transdisciplinary approach with the goal of achieving optimal health outcomes recognizing the interconnection between people, animals, plants, and their shared environment [19,20].

In the development and implementation of these strategies, physicians, patients, pharmacists, and veterinarians all play a key role. Identification of the non-clinical factors associated with the prescription of ABXs in animals is an essential step towards being able to implement effective interventions to improve ABX use in animals, and thereby slow the expansion of multi-drug resistant microorganisms worldwide. These factors have already been described in the case of physicians [21,22] and pharmacists [23] and it has been demonstrated that they can influence the use of antibiotics by these professionals, but to our knowledge there is no systematic review that identifies them in veterinarians. Despite being completely different sectors, we consider that, as occurs in human medicine, there may be factors other than clinical semiology itself affecting the prescription of antibiotics in veterinary medicine. Therefore, our designated aim was to conduct a systematic review to identify those factors and attitudes which do not form a part of clinical semiology or are the direct result of complementary tests, but which might influence the prescribing behavior on the part of veterinarians.

## 2. Results

### 2.1. Search Results

A total of 1386 papers were found in Medline, 1166 in Web of Knowledge, and 3209 in Commonwealth Agricultural Bureaux (CAB) Abstracts. The elimination of duplicates yielded 4570 studies. After reading the abstracts, 117 were selected as potentially valid, with 15 more papers being added on the basis of a review of the references cited in the original studies. Finally, after an in-depth reading of the full text, 34 studies were included in this review (see Figure 1. Flow chart).

### 2.2. Study Quality

All the studies of a quantitative or mixed nature complied with most of the criteria in the AXIS tool [24], and were thus deemed eligible for inclusion in our review. Whereas, 75% complied with almost all the items and were therefore deemed to display high methodological quality and low susceptibility to bias. The remainder complied with 16 to 17 items and were therefore considered to have a medium level of methodological quality (see Appendix A).

As shown in Appendix A, all the qualitative studies selected complied with the criteria contained in the CASP tool (exploratory questions 1 and 2 in the chart) [25], and in the remaining exploratory questions posed by the test. 

### 2.3. Characteristics of the Selected Studies

The general characteristics of the studies selected are summarized in Table 1.

Of the selected studies, nineteen were of a quantitative nature, fourteen of a qualitative nature, and one contained an analysis of both types. All were observational cross-sectional studies.

Twenty-two studies focused on industrial animal production and six on pets, whereas the other six were not confined to any specific sector.

Of the studies conducted in the industrial animal production, five were exclusively devoted to milk yielding, five to the pig industry, and one to beef production. The remaining studies did not target any specific area of production. 

In terms of pathology, four of the studies undertaken in dairy farming focused on udder health, and one focused on the prescription of ABXs against pig intestinal diseases. The rest of the studies did not limit their research to any specific clinical condition.

Most of the studies were from Europe, four from Oceania, three from Asia, two from America, and one from Africa.

Lastly, with respect to the methods used for data collection, most of the studies used semi-structured postal/e-mail questionnaires, face-to-face interviews, or focus discussion groups.

### 2.4. Factors Identified by the Selected Studies as Exerting an Influence

Each of the studies selected was examined to identify the non-clinical factors that might influence antibiotic prescribing by veterinarians. To facilitate classification, these factors were divided into two major blocks, intrinsic and extrinsic.

### 2.5. Intrinsic Factors

The factors intrinsic to professional veterinarians (summarized in Table 2) were further subdivided into socio-demographic characteristics, attitudes, etc.

#### 2.5.1. Socio-Demographic Characteristics

Among the studies of a quantitative nature, 10 evaluated the socio-demographic characteristics of veterinarians in relation to antibiotic prescribing. Gender did not prove statistically significant in four [31,35,37,40] out of the five studies [39], in which it was evaluated, with similar results being reported for age (three significant/four evaluated) [31,37,39,40] and professional status (3/5) [31,37,39,43,44]. In terms of professional experience, the results were inconclusive: While two studies observed greater professional experience to be an influence that apparently favored prescribing [39,40], one found the contrary [35], and four further studies reported that it was not a significant factor [31,36,41,43].

Professional training was evaluated by only two studies [35,40], without conclusive results being obtained.

#### 2.5.2. Attitudes

Most (31) of the studies in this review reported that certain attitudes displayed by veterinarians influenced their decision to prescribe antimicrobials. 

Fear of possible complications in the animal(s) as a result of not administering antibiotics was identified in eight quantitative studies [29,31,32,33,35,38,43], one of which objectively established a statistically significant relationship [42], and was cited as influential by eleven further qualitative studies [45,47,48,49,50,51,52,53,55,56,57]. Fear of losing customers as a result of failing to prescribe antibiotics was cited as having an influence by as many as nine studies [39,45,46,47,49,53,56,58,59], though in another study that used the statistical hypothesis test to evaluate the relationship, this did not prove significant [31].

Business factors were cited by nineteen studies [45,46,47,48,49,50,52,53,54,56,58,59], seven of which were of a quantitative nature [27,38,39,41,42,43,44]. In another study, statistical hypothesis testing failed to detect a relationship [40].

The confidence of veterinarians in their own prescribing habits, based on their prior experience, was highlighted as a significant factor when deciding whether to use antibiotics in 19 of the studies [26,28,29,32,33,35,36,39,42,43,46,47,48,49,52,53,56,57,58].

In 19 of the studies, we have identified that there could be complacency on the part of veterinarians with the expectations that they believe their clients have, and that this can have an influence when prescribing an antibiotic treatment [28,32,39,43,44,45,47,48,49,50,52,53,56,57,58,59].

Three quantitative studies [29,33,41] pinpointed the responsibility of other professional groups, such as physicians or pharmacists, as being the main cause of the spread of resistant strains. This was also described as a decisive factor by eight further studies of a qualitative [45,47,50,53,54,55,56] nature and one of a mixed nature [58].

Indifference was identified in seven studies [47,48,49,51,52,57,58]. 

Finally, the lack of appropriate knowledge of prescribing and antibiotic resistance was only mentioned by two studies [37,51].

#### 2.5.3. Other Intrinsic Factors

In studies that straddled more than one territory, veterinarians’ country, region, or origin did not seem to modify their intentions regarding prescribing [31,40,43]. However, this refers to several countries in the same area.

Three studies highlighted the fact that pressure exerted by more experienced colleagues could facilitate or decrease the prescription of ABXs [28,53,57], and one found that colleagues’ recommendations were a decisive influence when it came to the choice of antibiotic [26].

### 2.6. Extrinsic Factors

The extrinsic factors of the studies selected are summarized in Table 3. These were further subdivided into four sections:

#### 2.6.1. Customer-Related Factors (Farmer/Pet Owner)

As many as 16 studies reported that customers’ lack of knowledge and indifference to ABX use as well as the generation of resistance were factors that could boost ABX prescribing [27,29,32,35,41,45,46,48,50,51,52,53,54,55,56,59]. Connected with this, the high demand for antibiotics by customers was cited in 12 studies [34,37,39,41,43,47,48,53,55,56,57,58]. Additionally, in 12 other studies, veterinarians argued that customers’ lack of training in how to administer certain antibiotic guidelines or alternative therapies could properly compromise their decisions [38,39,43,45,47,48,52,53,54,56,57,59].

#### 2.6.2. Structural and Concurrent Factors

Grouped below are all the factors that determine the specific conditions under which veterinarians work at any given point in time.

In eight of the studies, the presence of inadequate hygiene and biosafety conditions on farms and agricultural holdings was highlighted as a factor that increased animals’ need for antibiotics [26,38,50,51,52,54,55,56]. For example, the lack of a suitable isolation circuit for sick animals or inadequate climatization systems. It should be highlighted that these types of factors were stressed by veterinarians working in the livestock sector, not by those working with household pets.

Many veterinarians also cited the lack of time for examining animals properly and the low number of veterinarians per clinic as factors that influenced treatment decisions [28,29,47,57,58]. Two studies evaluated whether having two or more veterinarians per clinic served to improve antibiotic prescribing, without significant results being obtained [31,43]. 

Three studies underscored the fact that the pharmaceutical industry can alter veterinarians’ antibiotic prescribing in different ways [34,44,48].

Many studies singled out the long time [26,29,30,39,42,45,47,56] and high price [33,38,42,50,56,59] involved in obtaining the results of culture and antibiogram sensitivity tests as important bars to the judicious prescription of ABXs.

With respect to antibiotic policy, four studies reported that the restrictions imposed by the health authorities in different European countries on the administration of certain antibiotics to animals served to reduce ABX misuse [26,42,46,47]. Furthermore, the absence of animal-centered ABX prescription policies was mentioned by six other studies as a factor that facilitated inappropriate prescribing [34,39,50,51,56,58].

Prescription guidelines were perceived as positive by almost all veterinarians, but their lack of clarity was highlighted by one study [29].

#### 2.6.3. Antibiotic-Related Factors

Nine studies indicated that antibiotic prescribing was often influenced by the unavailability of appropriate antimicrobials or the impossibility of applying different therapeutic alternatives to an antimicrobial treatment (due to the lack of material, time, or training) [34,35,47,48,50,53,55,57,59]. The cost of some ABXs was also perceived as a bar by eight other studies [26,29,34,36,38,43,47,55]. These limitations can make veterinarians lean towards higher importance antimicrobials than recommended in the guidelines.

A number of studies reported that, aside from their spectrum of action, certain characteristics of the antimicrobials available on the market exert an influence when it comes to choosing a treatment. 

Up to nine studies concluded that the dosage and form of administration [26,29,35,36,42,43,44,47,57] of the available antimicrobials are decisive factors when deciding what to prescribe. Therefore, making antimicrobials which require subcutaneous injections and fewer doses, the preferred choice of many veterinarians: One example of this is cefovecin, an antibiotic chosen to treat a large number of cat and dog diseases due to its subcutaneous route of administration and just one injection in most cases.

Another aspect linked to the characteristics of antibiotics available in the veterinary sector is what is known as the withdrawal (or withholding) period. This is the mandatory time which must elapse between the last application of an antibiotic and use of animal-based products and is aimed at ensuring that there are no antibiotic residues in these products or that they lie within the permitted limits for the medication and foodstuff in question. This period varies, depending on the antibiotic and the livestock species in which it is used, the animal product to be used, and the jurisdiction and consequences for non-compliance are severe. This factor was cited by veterinarians as fundamental in as many as six studies [34,38,39,43,45,55].

#### 2.6.4. Animal-Related Factors

The clinical profiles exhibited by animals (signs, results of complementary tests, etc.), are the most influential animal-related factors, when it came to deciding whether or not an antibiotic was to be prescribed, did not come within this review’s designated scope of analysis. 

Clinical aspects aside, few animal-related factors were identified. In five studies [38,45,46,55,56], veterinarians maintained that animals produced by intensive animal breeding displayed a weak immune system, which rendered them more vulnerable to infections and, by extension, to receiving anti-biotherapy. In two other studies [47,57], veterinarians stated that the animal’s behavior, especially if aggressive, could modify the treatment decision by favoring the administration of broad-spectrum injectable antibiotics or oral medication if it is not possible to approach the animal.

The remaining factors mentioned in the studies were cited by few papers or yielded widely varying results.

## 3. Discussion

To our knowledge, this is the first review to analyze the various factors, other than purely clinical aspects, that can influence veterinarians when it comes to prescribing antibiotics. Different attitudes of veterinarians, such as fear, self-confidence, and complacency, can influence prescribing, as can extrinsic factors, ranging from poor or non-existent customer awareness with antibiotic stewardship to inadequate farm hygiene and lack of regulation, along with certain characteristics of available ABXs, such as being cheap or easy to administer, and the rapidity and cost of culture and sensitivity tests. Many of these factors are potentially modifiable and could well be the focus of future interventions targeted at improving ABX use in animals.

### 3.1. Socio-Demographic Factors

It appears that neither the veterinarians’ gender, age, nor professional status influences prescribing. The only socio-demographic characteristic that seems to exert an influence is professional experience: A number of studies suggested that less experienced veterinarians could favor more ABX prescribing, partly due to the fact that they lack confidence in their diagnostic skills and are thus more vulnerable to the opinions of third-parties such as clients. Clinical practice protocols and prescription guidelines can help veterinarians acquire greater confidence [37,41,42,48,53,57,58]. Indeed, veterinarians who make use of prescription guidelines, scientific literature, and the advice of specialists are less inclined to prescribe incorrect dosages than those who do not use these data sources [44]. Other studies, in contrast, indicated that it is the most experienced veterinarians who have a greater tendency to prescribe antibiotics: This may be due to the fact that their years of training lie in the past, at a time when far less stress was laid on the prudent use of ABXs. Consequently, even veterinarians with decades of experience could benefit from refresher courses to update them on the subject [30,39,57,58]. Even so, recent studies report a substantial variation in compliance with treatment guidelines or accepted standards among Australian veterinarians, depending on the species treated and the clinical circumstances. These same studies also show that socio-demographic characteristics do not influence compliance with guidelines, and the results are similarly contradictory with respect to veterinarians’ year of graduation [60,61,62,63]. No relationship has been found between these results and the region or professional sector in which the study was contextualized. 

### 3.2. Attitudes

Some of the veterinarians’ attitudes do indeed appear to influence prescribing, such as confidence in their own decisions, influenced by their habits and previous experiences. Previous experience is a useful tool when taking decisions but the excess of trust in customs and experience may act as a barrier to correct prescribing and hinder the incorporation of new information. To minimize this, among other measures, we believe it is important to ensure access to up-to-date scientific information through continuing education [30,33,43,44,47,49,50,58], whether via face-to-face courses or by taking advantage of currently available digital resources, with an emphasis at all times on the importance of prudent ABX use and the negative consequences of misprescription [42,54,56,57]. It would also be useful to implement clinical decision-support systems to increase these practitioners’ confidence [28].

Many studies make the point that both veterinarians’ explicit and implicit client expectations of antibiotics can lead to misprescription, something that has also been reported by physicians and pharmacists [22,23]. For example, some veterinarians stated that they felt their customers expected something tangible from a consultation, in the form of medication [48]. Therefore, veterinarians must be empowered by the health authorities, in order to ensure that they feel responsible for taking the lead in the rational use of ABXs, and that this criterion prevails over external pressures [48,57]. 

Many veterinarians fear the possible negative consequences of not prescribing ABXs for an animal’s wellbeing at a given point in time. This same fear has been shown to be decisive in prescribing decisions in the sphere of human medicine [21,23]. The perception of risk and diagnostic uncertainty is a known driver of human behavior [64,65].

In order to reduce dependence on antibiotics and to avoid the consequences which a selection of resistant strains may have for individual animals or for a farm as a whole, preventive measures are essential [38,39,41,56], for example, good management, good feeding, regular veterinary controls to ensure vaccination and parasite treatments, good internal and external biosecurity, etc. [32,36,39,49,53,55,57].

Sight should not be lost on the fact that veterinary practice is often framed within the activities of the private sector, in which veterinarians acknowledge how different business factors, such as customers’ economic possibilities [51,57], the price of preventive interventions or alternative therapies [54], indirect costs, market tensions, competition [59], etc., can influence their prescribing. However, we never perceive the quest for personal economic benefit. Likewise, and closely linked to the above factors, the fear of losing a customer as a result of not dispensing an ABX (something that also happens with pharmacists [23]), was also highlighted [56,59].

It has also been suggested that there is a conflict of interest between correct prescribing and the profit that veterinarians themselves can obtain from the direct sale of ABXs [44,53]. For this reason, it has been proposed that prescribing be “decoupled” from dispensing [66]. However, the majority of veterinarians reject the notion that the prospect of profits would encourage the inappropriate sale of ABXs [32,38,55,56,58] and oppose any such separation [49,53,56]. 

Similar to some physicians [21] and pharmacists [23], a limited number of veterinarians maintain that the generation of ABX resistance is the responsibility of other professional groups, whether involved in human medicine or other veterinary sectors. Although a few professionals even question the contention that ABX use in animals could pose a real threat to human health [54,55], most veterinarians showed a high level of awareness and admitted to being extremely concerned about the spread of resistance [26,34,36,55]. Efforts must continue to be made to raise awareness among professionals in the sector [48,49,54,56].

Some veterinarians believe that a 50% reduction in ABX use on farms is feasible [38,40]. However, to achieve this, a close collaboration between veterinarians and customers [40], and adequate communication skills are both fundamental [38,48].

### 3.3. Client-Related Factors

Many veterinarians identified a strong customer demand for ABXs as a factor that could influence their treatment decisions. This is similar to factors identified as determinants in human medicine [21], and is closely related to customers’ lack of knowledge and indifference to the misuse of ABXs and spread of resistance. These results, obtained in human and animal health, reflect the need to enhance the general population’s education and awareness regarding ABX use, which would bring benefits across the sector [32,37,38,39]. In fact, many veterinarians in Europe stressed that customer awareness had improved in recent years [48], which indicates that the educational interventions for the general population that are being carried out could be yielding results [30,38,56], as has also been seen in human health. Little evidence has been found of similar shifts in public sentiment towards a more careful antibiotic use in developing countries, so public educational interventions could be helpful in these settings [50].

At present, and especially in high income-countries, there is a growing consumer pressure to reduce ABX use on farms. Although this might serve to reduce farmer requests for antibiotics, there is a danger of creating demand for animals being raised without any antibiotics at all, which can lead to a negative animal health and welfare outcome [49,54].

### 3.4. Structural and Concurrent Factors

Of the different interventions to reduce ABX use, those aimed at improving biosafety and hygiene were pinpointed as some of the most important [30,32,34,36,37,38,40,41,53,55,56]. These measures are indispensable for reducing the consumption of antibiotics, as has been shown by several field studies [67,68,69]. It should not be forgotten, however, that in an industry where the profit margins are very narrow, many of these interventions are very expensive [53], thus leading to the proposal that economic incentives should be provided to enable farmers to apply them [50]. 

With respect to the approach adopted by antibiotic policies, two perspectives are to be found: (1) Endeavoring to improve ABX use through voluntary initiatives and programmes promoting responsible ABX use [32,40,48], or alternatively, (2) opting for more interventionist policies, as in Scandinavian countries [30,34,50]. Some countries also use a mix: Voluntary initiatives promoting responsible use with government support and some policy changes in areas where needed.

Despite the good results obtained in Sweden and Denmark in terms of reducing ABX use in animals [13], it is difficult to ascertain whether this is a direct consequence of their more tightly controlled ABX policies, e.g., Denmark’s introduction of ABX-use thresholds for farms and fines for exceeding these [26,70]. Furthermore, the data furnished by other countries, such as the United Kingdom, Germany or Belgium, with their promotion and awareness-raising policies, are also good [32]. In general, veterinarians display reticence towards these more restrictive policies, since they perceive them as a loss of professional autonomy [33,55] and fear the possibility of being unfairly fined [31,37,47]. 

Regardless of the debate about the degree of intervention by the authorities, legislative measures should foster prudent ABX use in all sectors (human, animal, and environmental) [34]. Many veterinarians are in favor of reinforcing the regulatory framework [50,51,54,58]. There are many measures that can be sponsored by health and government institutions, e.g., strengthening infectious-disease surveillance systems [38,41], providing databases on ABX resistance at a local and national level [36], restricting the sale of ABXs without a previous formal veterinary consultation [58], prescription monitoring with indicators and feedback to veterinarians [34,50], implementing electronic prescription [33], introducing comparative prescription-evaluation systems [38,41], limiting the prescription of certain ABXs [49], etc. All this poses a challenge for the authorities, and an effort should therefore be made to ensure that any regulations which are introduced are not perceived by professionals as merely disciplinary [33]. The authorities should recognize the enormous effort made by the sector over recent years in many regions, which has led to a drastic reduction in the use of antibiotics, and support veterinarians in such a way that they can continue leading this task. 

In some studies, professionals complained that the lack of time available for care and the lack of veterinarians might lead to misprescription. Even though we know that veterinary practices are private businesses with generally low margins in a competitive business environment, minimum times for proper health care management could be standardized and respected [57]. 

In many cases, pharmaceutical companies not only constitute one of the data sources for veterinarians [43,47], but also dictate existing ABX product lines [47,49]. The role of the industry in developing countries has proved more conflictive, since there have been reports that, in certain settings, companies sold and advertised their products directly to farmers without prescription or veterinary advice [50]. Of the selected studies, only five were set in the context of low to middle-income countries. The situation in these countries with regard to the use of ABX in comparison to that of countries in Europe and North America and Australia and New Zealand (in which the rest of the studies took place) is very different. In these five studies, as in different international reports [71], great concern is shown for the inappropriate use of antibiotics in these regions, with lack of access to laboratory tests [50], poor training [37], negligible awareness [51], absence of regulation [37], selling antibiotics directly to consumers without prescription, etc. Accordingly, in such countries, where an important part of worldwide animal production is concentrated [9], interventions targeted both at enhancing awareness and training, and at strengthening the regulatory framework are especially necessary [37,50,51].

### 3.5. Culture and Antibiogram Test

One of the recommendations and guidelines for rational ABX use in animals is to confirm the diagnoses with the aid of culture and antibiogram tests. Although some EU countries have now made the use of AST before using CIA’s obligatory, a number of studies show that the use of these laboratory tests as a decision-making support tool is not as widespread among professionals as it should be [36,42,44,48,51]. Some veterinarians admit that they only resort to them in the event of recurrent infections or treatment failure [32,51], and we have noted in the studies that the cost and waiting time involved in obtaining results are the main reasons for their use being limited. In addition, the difficulty of justifying their use to clients must be added, when the latter demand the fastest and cheapest solution possible [57]. Hence, the development of cheaper, faster, and more reliable tests might increase their use and decrease the misuse of ABXs [33,39,53,56]. Furthermore, veterinarians’ and livestock farmers’ awareness of the importance of such tests need to be enhanced, and health authorities could assess the desirability of subsidizing their cost, wholly or in part [51].

### 3.6. Antibiotic-Related Factors

Some characteristics of ABX pharmacokinetics (route of administration, duration of ABX activity, and time of elimination) are influential when it comes to choosing the treatment, and are closely related with two other factors, namely, the skill of the farmer or owner in applying the indicated treatment and the animal’s co-operation. This has led many veterinarians to prefer injectable formulations with long-acting antibiotic preparations as a means of ensuring therapeutic compliance: An example of this would be cefovecin. If the appropriate antibiotic is not available for each animal and for each pathology with an easily applicable route of administration, other, less suitable, antibiotics may be employed, increasing the risk of generating resistance. The problem is that the WHO and OIE regard cefovecin and other antimicrobials, as critically important for human and animal health, and recommend that their use should be restricted [17]. This highlights how the lack of availability of given antibiotics or certain presentations can cause other ABXs to be prescribed, in the knowledge that they are not the best medicines indicated [47,51,55]. Work should be done to develop new pharmacological presentations that would enable veterinarians to treat all the animals on the principle of antimicrobial stewardship, e.g., injectable long-acting formulations of antibiotics of lower importance, with an effective duration of action of a few days [51,58]. 

Another important factor for veterinarians working in the food sector is the withdrawal period. This is due, among other reasons, to the fact that the use of drugs with a short withdrawal period in animals reduces costs, since the relevant animal-based product(s) can be brought onto the market that much sooner. Hence, health authorities should try to take into account this limitation when drawing up prudent ABX prescription protocols, and envisage more than one alternative treatment for each situation, although we are aware that the alternatives are very limited by the available and effective antibiotics for each species in each country [51,53].

### 3.7. Animal-Related Factors

Several European studies reported that some veterinarians believed that livestock in intensive production systems had poor immunity. It is important to monitor the health of the herd [36] and also to ensure the quality of the animal feed [38,41] and promote vaccination [30,32,37,53,55]. 

### 3.8. Alternative Treatments

Some veterinary conditions where antibiotics are often applied could be managed solely with non-antibiotic treatments, such as surgical drainage and non-steroidal anti-inflammatory drugs for abscesses treatments [38,41]. While many veterinarians are in favor of these [31], many others are skeptical about their efficacy and clients’ ability to apply them [52]. Therefore, it is important to invest in their promotion and in more clinical trial evidence to convince both veterinarians and animal owners that outcomes are equivalent without antibiotics [30,52].

It should be noted that all the interventions proposed are useful for trying to minimize the influence which many of the factors identified in this review can exert on the decision to prescribe, and thus it is essential that all sectors implicated co-operate fully in their implementation.

### 3.9. Limitations

The main limitation of this review lies in that we included studies embedded in both the companion and animal-production settings, although fully aware that they are very different sectors.

It should also be mentioned that even though the regulatory framework is very different across countries, we have not differentiated between the results achieved by the geographical area (country). Along this line, almost all (reviewed) studies are European-based, where the use of antibiotics is highly controlled, with almost no studies found in developing countries, where a substantial fraction of ABX usage in animal production is concentrated. Furthermore, the studies’ results have not been interpreted according to the date in which the data were collected, although it is well known that the legal framework and restrictions have changed considerably in recent years.

When interpreting the results, we must also take into account the methodological heterogeneity of the studies selected, particularly in terms of data analysis, scales used, and the fact that not all the studies focused on the analysis of the same variables. Furthermore, the results have been collected according to the number of studies in which each factor is mentioned, although there are significant variations between the number of veterinarians participating in each study. Consequently, the results of this review should be interpreted with caution. In our understanding, the abovementioned heterogeneity in methods, results, and measurements precludes us from the possibility of performing a meta-analysis.

All things considered, we maintain that these limitations in no way alter the main conclusions reached. There is a great consistency found in the results in different settings. In addition, they are analogous to those obtained in physicians [21,22] and pharmacists [23].

Identifying these factors is an essential step to optimize the prescription of antibiotics in animal settings.

Finally, we consider that the relationship between the identified factors and the prescription of ABXs will have to be further detailed in the future by quantitative observational studies. Therefore, making the development of validated and reliable instruments (e.g., questionnaires) indispensable to obtain results with a higher degree of evidence.

## 4. Materials and Methods

This study was conducted in accordance with the Preferred Reporting Items for Systematic Reviews and Meta-Analyses (PRISMA) guidelines. The standards of the study were ensured with the aid of the PRISMA 2009 checklist (Appendix A), PROSPERO registration number: CRD42020189747.

### 4.1. Search Strategy

To achieve the designated goals of this systematic review, a search was made in the Medline, Web of Knowledge and Commonwealth Agricultural Bureaux (CAB) Abstracts databases, covering an unlimited time period until March 2020, and using the following search terms: (Veterinarian OR veterinary) AND (antibiotic* OR antimicrobial*) AND (factor* OR attitude* OR knowle* OR percept* OR perceiv* OR belief*) AND (use OR misuse OR overuse OR practice* OR prescription).

### 4.2. Inclusion and Exclusion Criteria

The papers included were required to meet the following inclusion criteria: Studies were published in English or Spanish, and designed to examine factors and attitudes associated with ABX prescribed by veterinarians, regardless of the sector in which they carry out their activity (pets, horse industry, animal production). The studies included used both quantitative and qualitative methodologies.

### 4.3. Quality Assessment

When it came to evaluating the suitability of the studies selected, the two following tools were used to assess the quality and susceptibility to bias: In the case of qualitative studies, the Critical Appraisal Skills Programme (CASP) tool [25], and in the case of quantitative and mixed studies, the critical appraisal tool to assess the quality of cross-sectional studies (AXIS) [24]. An important aspect of these tools is that they evaluate whether the conclusions reported by the studies are both credible and consistent with respect to their stated objectives, methods, analysis, and results.

### 4.4. Data-Extraction and -Analysis

All data were extracted from the selected studies, bearing in mind the methodological nature of the original study. Two of the authors (MS-D and M.T-T) reviewed the studies separately, with any disagreements being settled by consensus.

A first table (see Table 1) was drawn up to summarize the methodological characteristics of the papers selected.

Each of the studies selected was examined to identify the factors and attitudes that might influence veterinarians when it came to prescribing antibiotics. To attain the review’s stated goal, data were not collected on factors relating to clinical issues (e.g., clinical signs manifested by the sick animal) or the results of complementary tests likely to be performed on animals susceptible to diseases of infectious origin (e.g., imaging tests, laboratory results, etc.).

To facilitate the interpretation of results, factors identified as determinants were divided into two categories, namely, factors intrinsic and extrinsic to the veterinary professional.

The intrinsic factors were, in turn, subdivided into two blocks, i.e., socio-demographic characteristics and attitudes. The socio-demographic characteristics analyzed in the studies were age, gender, years of experience, professional status, and professional training.

To ensure the correct allocation, prior to extracting the data, and based on the results of similar studies undertaken by our group [22,23], a series of attitudes were predefined as potential influences when veterinarians came to decide whether or not to prescribe an antibiotic. These were:Complacency (with client expectations): The attitude that causes antibiotics to be prescribed in order to meet what veterinarians perceive as their clients’ expectations.Fear: The attitude that reflects the fear felt by veterinarians at the prospect of the future medical complications which an animal might suffer (Fear 1) or of the loss of a customer (Fear 2), as a consequence of not prescribing an antibiotic at a given point in time.Responsibility of others: The attitude which reflects the belief held by veterinarians that the responsibility for the generation of antibiotic resistance in the community is attributable to other professional groups.Self-confidence: The level of confidence felt by practitioners when it comes to prescribing an antibiotic, backed by their own habits and previous experience.Business factors: The set of attitudes that reflects the tendency to prioritize financial and/or business factors when it comes to prescribing an antibiotic.Indifference: The absence of motivation to perceive, positively or negatively, the problems associated with the misuse of antibiotics and, by extension, with antibiotic resistance.Ignorance: The treatment prescribed by the veterinarians does not match the current knowledge and recommendations for treating the problem in question. Therefore, it is a reflection of the veterinarians’ lack of proper knowledge.

Lastly, factors extrinsic to the veterinarian were analyzed and grouped into four categories:Customer-related factors (farmer or animal owner).Structural and concurrent factors.Antibiotic-related factors.Animal-related factors.

For the purpose of evaluating the relationship between the above-mentioned factors and the antibiotics prescribed in papers that furnished the quantitative results, the following terms were defined:Statistically significant relationship: In the studies reviewed, the presence of this factor increased the ABXs prescribing by veterinarians. This was deemed statistically significant when the measure of association reported in the primary study was higher than 1 and was associated with a *p*-value ≤0.05. This was denoted in the tables as (↑).Statistically significant inverse relationship: In the studies reviewed, the presence of this factor decreased the ABXs prescribing. This was deemed statistically significant when the measure of the association reported in the primary study was lower than 1 and was associated with a *p*-value ≤0.05. This was denoted in the tables as (↓).No significant relationship: In the studies reviewed, the presence of this factor had no influence on the ABXs prescribing. This was deemed to happen when the measure of the association reported in the primary study was associated with a *p*-value > 0.05, which was not statistically significant. This was denoted in the tables as (≠).No symbol: In the study reviewed, the factor indicated was related with the ABXs prescribing, but this relationship was not evaluated by statistical hypothesis testing.

## 5. Conclusions

As part of the overall strategy for reducing the spread of antibiotic-resistant bacterial strains worldwide, the excessive and inappropriate use of antibiotics must also be reduced in animals, and in this respect, veterinarians are a key element. Our results suggest non-clinical factors which are associated with veterinary antibiotic prescribing (e.g., self-confidence, biosafety deficits, fear, lack of awareness, inappropriate policies, etc.). These are potentially modifiable, and could thus be the target of future interventions designed to optimize ABX use in this sector, and decrease the prevalence of resistant bacteria in animals and human beings alike.

## Figures and Tables

**Figure 1 antibiotics-10-00133-f001:**
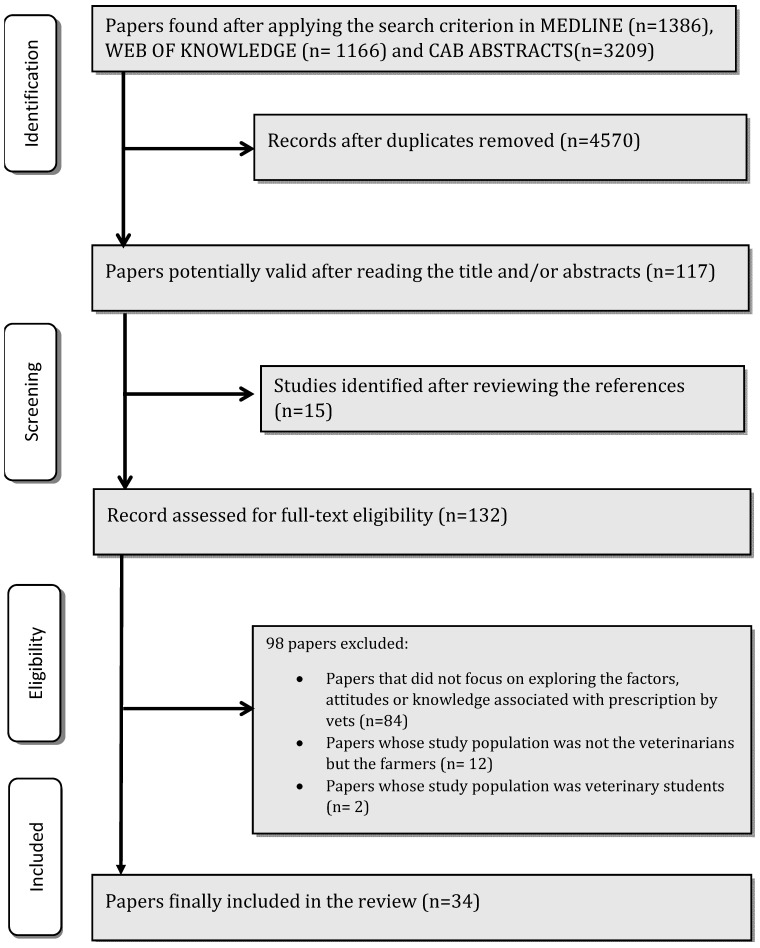
Flow chart.

**Table 1 antibiotics-10-00133-t001:** Methodological characteristics of the papers selected.

Author	Year	Country ^a^	Study Population ^b^	Veterinary Practice Type	N ^c^	% R ^d^	Disease	Data-Collection ^e^
**Quantitative**								
Eriksen [26]	2019	Denmark	veterinarians	livestock (swine)	105	83	Intestinal diseases	Q
Hopman [27]	2019	Netherlands	veterinarians	small companion animal	350	22		Q
Doidge [28]	2019	UK	veterinarians	livestock	306	24		Q
Norris [29]	2019	Australia	veterinarians	multiple	403	4		Q
Carmo [30]	2018	EU	veterinarians	livestock	67	-		Q
Scherpenzeel [31]	2018	Netherlands	veterinarians	livestock (dairy cattle)	181	28	Udder health	Q
Coyne [32]	2018	UK	surgeons veterinarians	livestock (swine)	61	34		Q
Zhuo [33]	2018	Australia	mixed	multiple	403	4		Q
Kumar [34]	2018	India	veterinarians	livestock	48	-		Q
Ekakoro [35]	2018	USA	veterinarians	multiple	62	51		Q
Barbarossa [36]	2017	Italy	veterinarians	small companion animal	266	12		Q
Anyanwu [37]	2017	Nigeria	vet	livestock	280	76		Q
Postma [38]	2016	EU	veterinarians	livestock	728	33		Q
Mc Dougall [39]	2016	New Zealand	mixed	livestock (dairy cattle)	206	34	Udder health	Q
Visschers [40]	2015	EU	mixed	livestock (swine)	334	-		Q
Speksnijder [41]	2015	Netherlands	vet	livestock	377	34		Q
De Briyne [42]	2013	EU	veterinarians	multiple	3004	1,5		Q
Gibbons [43]	2012	Ireland	veterinarians	livestock (cattle)	118	66		Q
Hughes [44]	2011	UK	veterinarians	small companion animal	460	51		Q
**Qualitative**								
Golding [45]	2019	UK	mixed	livestock	13	-		Telephone interview
Pucken [46]	2019	Switzerland	veterinarians	livestock (dairy cattle)	23	-	Udder health	FG
Hopman [47]	2018	Netherlands	veterinarians	multiple	18	72		Face to face
Smith [48]	2018	UK	veterinarians	small companion animal	16	-		Face to face
King [49]	2018	UK	surgeons veterinarians	small companion animal	16	-		Face to face
Chauhan [50]	2018	India	mixed	livestock	9	-		Face to face
OM [51]	2017	Cambodia	mixed	livestock	9	-		Face to face
Higgins [52]	2016	UK	veterinarians	livestock (dairy cattle)	20	100	Udder health	Face to face
Coyne [53]	2016	UK	surgeons veterinarians	livestock (swine)	21	-		Face to face
Etienne [54]	2016	EU	mixed	livestock	30	16		Face to face
Coyne [55]	2014	UK	veterinarians	livestock (swine)	9	-		FG
Speksnijder [56]	2014	Netherlands	veterinarians	livestock	11	88		Face to face
Redding [59]	2014	Peru	mixed	livestock (dairy cattle)	12	-		FG
Mateus [57]	2014	UK	veterinarians	small companion animal	21	-		Face to face
**Mixed**								
Hardefeldt [58]	2018	Australia	veterinarians	multiple	184	-	Q	Q

^a^ Country: EU: Two or more countries of the European Union; UK: United Kingdom. ^b^ Study population: Mixed, several study populations, only veterinary results were collected. ^c^ Sample size: Number of veterinarians that were included in the study. ^d^ Percentage response: Percentage of veterinarians on whom the study could be successfully carried out. -: Unknown. ^e^ Data collection: Q: Questionnaire; FG: Focus group; Face-to-face: Qualitative face-to-face interview of a semi structured nature.

**Table 2 antibiotics-10-00133-t002:** Intrinsic factors identified as influencing antibiotic prescription by veterinarians.

Author	Socio-Demographic Characteristics ^a^					Attitudes ^b^								Others
	Age	Gender	Professional Status	Experience	T	I	Complacency	Fear 1	Fear 2	Responsibility of Others	Business Factors	Confidence	ID	
**Quantitative**														
Eriksen [26]												C		Advice from colleagues
Hopman [27]											BF			
Doidge [28]	<30 ↑						CO ↑					C ↑		vet practice type(farm) ↑
Norris [29]								F1		RO		C		
Carmo [30]														
Scherpenzeel [31]	A ≅	G ≅	PS ≅	E ≅				F1						Region ≅
Coyne [32]							CO	F1				C		
Zhuo [33]								F1		RO		C		
Kumar [34]														
Ekakoro [35]		G ≅		>E ↓	T≅			F1				C		Veterinary practice type ≅
Barbarossa [36]				E ≅								C		
Anyanwu [37]	A ≅	G ≅	PS ≅			I								
Postma [38]								F1			BF			
Mc Dougall [39]		Male ↑	Owner ↑	>E ↑			CO		F2		BF	C		
Visschers [40]	A ≅	≅		>E ↑	T↑			F1 ↑	F2 ≅		BF ≅			Country ≅
Speksnijder [41]				E ≅						RO	BF			
De Briyne [42]											BF	C		practice type(equine) ↓
Gibbons [43]			PS ≅	E ≅			CO	F1				C		Country ≅
Hughes [44]			Locums ↑				CO				BF			
**Qualitative**														
Golding [45]							CO	F1	F2	RO	BF			
Pucken [46]									F2		BF	C		
Hopman [47]	Old						CO	F1	F2	RO	BE	C	ID	
Smith [48]	young						CO	F1			BF	C	ID	
King [49]							CO	F1	F2		BF	C	ID	
Chauhan [50]							CO	F1		RO	BF			
OM [51]						I		F1					ID	
Higgins [52]				<E			CO	F1		RO	BF	C	ID	
Coyne [53]							CO	F1	F2	RO	BF	C		Peer Pressure
Etienne [54]										RO	BF			
Coyne [53]								F1		RO				
Speksnijder [56]							CO	F1	F2	RO	BF	C		
Redding [59]							CO		F2		BF			
Mateus [57]							CO	F1				C	ID	Peer Pressure
**Mixed**														
Hardefeld [58]							CO		F2	RO	BF	C	ID	One vet in practice

Statistical significance (only quantitative studies). ≅: This factor was not statistically significant; ↑: This factor leads to a statistically significant increase in antibiotic prescription; ↓: This factor leads to a statistically significant decrease in antibiotic prescription; No symbol: In the reviewed article, this factor was associated with prescription, but there was no hypothesis testing. ^a^ Socio-demographic characteristics: A: Age; G: Gender; PS: Professional status; E: Years of professional experience; T: Professional training courses; ^b^ Attitudes: I: Ignorance (lack of knowledge); C: Complacency; F: Fear; RO: Responsibility of others; BF: Business factors; CF: Confidence in previous experience and habits; ID: Indifference.

**Table 3 antibiotics-10-00133-t003:** Extrinsic factors identified as influencing antibiotic prescription by veterinarians.

Author	Client ^a^			Structural ^b^						Antibiotic ^c^				Animal ^d^		Others
	LK/A	LT	HD	BS	TP	PI	PCT	DCT	LR	AA	AC	Dosage	WP	I	AB	
**Quantitative**																
Eriksen [26]				BS					LR		AC	Dosage				
Hopman [27]	LK/A															
Doidge [28]					TP ↑											Economic problems client ↑
Norris [29]	LK/A				TP		PCT	DCT			AC	Dosage				Unclear prescription guidelines
Carmo [30]								DCT								Digestive and respiratory diseases
Scherpenzeel [31]																No. of vets in practice ≅
Coyne [32]	LK/A															
Zhuo [33]																
Kumar [34]			HD			PI			LR	AA	AC		WP			
Ekakoro [35]	LK/A									AA		Dosage				
Barbarossa [36]											AC	Dosage				
Anyanwu [37]			HD													
Postma [38]		LT		BS							AC		WP	I		
Mc Dougall [39]		LT	HD					DCT	LR				WP			
Visschers [40]																Good relationship with clients ↓
Speksnijder [41]	LK/A		HD													
De Briyne [42]							PCT	DCT	LR ↑			Dosage	WP≅			
Gibbons [45]		LT	HD								AC	Dosage	WP			No. of vets in practice ≅
Hughes [44]						PI ↑						Dosage				Referral hospital ↓/Acredited Hospital ↑
**Qualitative**																
Golding [45]	LK/A	LT						DCT					WP	I		Client’s Personality
Pucken [46]	LK/A								LR					I		
Hopman [47]		LT	HD		TP			DCT	LR	AA	AC	Dosage			AB	
Smith [48]	LK/A	LT	HD			PI	PCT	DCT		AA						
King [49]																
Chauhan [50]	LK/A			BS					LR	AA						Not enough veterinarians
OM [51]	LK/A			BS					LR							Lack of surveillance systems
Higgins [52]	LK/A	LT		BS												
Coyne [55]	LK/A	LT	HD							AA						
Etienne [54]	LK/A	LT		BS												Cultural barriers
Coyne [55]	LK/A		HD	BS						AA	AC		WP	I		Unprofessionalism
Speksnijder [56]	LK/A	LT	HD	BS			PCT	DCT	LR					I		
Redding [59]	LK/A	LT								AA						Different antibiotic packaging
Mateus [57]		LT	HD		TP		PCT			AA		Dosage			AB	Economic problems client
**Mixed**																
Hardefeldt [58]			HD		TP		PCT		LR							

Statistical significance (only quantitative studies) ≅: This factor was not statistically significant; ↑: This factor leads to a statistically significant increase in antibiotic prescription; ↓: This factor leads to a statistically significant decrease in antibiotic prescription; no symbol: In the reviewed article, this factor was associated with prescription, but there was no hypothesis testing. ^a^ Client-related factors: LK/A: Lack of knowledge or awareness; LA: Lack of training; HD: High client demand. ^b^ Structural: BS: Defects in biosecurity or hygiene on the farm; TP: Time pressure; PI: Pharmaceutical industry; PCT: Price of culture tests and sensitivity; DCT: Delay of culture tests and sensitivity; LR: Lack of regulation in antibiotic policy. ^c^ Antibiotic: AA: Antibiotics availability; AC: Antibiotic cost; D: Available antibiotic dosage; WP: Withdrawal period. ^d^ Animal characteristics: I: Immunity, understood as a defense mechanism against infection; AB: Animal behavior.

## Data Availability

No new data were created or analyzed in this study. Data sharing is not applicable in this article.

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
