# Peer review of "Non-Clinical Factors Determining the Prescription of Antibiotics by Veterinarians: A Systematic Review"

_antibiotics, 2021, doi:10.3390/antibiotics10020133_

Round 1
Reviewer 1 Report
This is an interesting meta-study on an important topic, what factors define the prescribing behaviour of veterinarians. It is based on only a limited number of original research papers which are themselves difficult to compare, as they are from different parts of the world and cover different types of veterinary practices e.g. companion animal versus pig or dairy practice.
My main problem is that the publication is making conclusions on studies that are different to compare. Some conclusions made I can not find back in the results. Often generalisations are used such as 'many found that xxxx', while the 'many' is not defined and often based on a result of 1 or 2 papers.
While the topic and intention are very good, I believe the paper needs to be very much rewritten to make it more scientifically sound and to make sure that the results and conclusions are really supported by the findings in the different papers.
I would also suggest breaking up some results for the different regions.
You can find many more detailed comments in the paper attached.

Reviewer 2 Report
The study entitled “Non-clinical factors that determine the prescription of antibiotics by veterinarians: A systematic Review” is a review of some factors other than clinical that can influence veterinarians decision to prescribing antibiotics. This is an interesting study to increase knowledge of the non-clinical considerations of prescribing antibiotics in veterinary. However, I have some questions and comments to Authors.
Abstract:
Please specify the main overall purpose of this study.
Introduction:
Lines 54-55: please include a few words about main assumptions of “One Health approach” on the basis of references cited.
Lines 56-57: please consider addition in this part of the text more precise description of potential non-clinical factors involved in prescription of ATB
Lines 60-61: please consider addition into this part the main similarities and differences between factors influencing decisions of ATB prescription in humans and animals.
Results:
Lines 183-185: please describe possible relation between ATB use and organization of farms. Are some data concerning use of ATB in farming systems such as agri-environment schemes or organic farms available? Did the authors observe any relation between types of practice (pets, horse industry, animal production) and evaluated indices?
Lines: 188-190: are there some specific factors for example involved in organization of veterinary clinics or farm specific?
Discussion:
Lines 270-275: due to the serious discrepancy in the results achieved, it may be worth clarifying whether the values depended on age were additionally related to the country or type of veterinary practice or other factors
Lines 412-425: Which part of results is discussed here? Please indicate.
Lines 427-440: please take into account and describe the question how the formulation of antibiotic and the route of administration influenced the misuse (as in case of ototopical ointments composed of several antibiotics).
Author Response
Response to Reviewer 2 Comments
Comments and Suggestions for Authors
The study entitled “Non-clinical factors that determine the prescription of antibiotics by veterinarians: A systematic Review” is a review of some factors other than clinical that can influence veterinarians decision to prescribing antibiotics. This is an interesting study to increase knowledge of the non-clinical considerations of prescribing antibiotics in veterinary. However, I have some questions and comments to Authors.
Response: Many thanks for having taken the time to read our paper. We are sure that your interesting comments have led to its significant improvement. Below, we present our responses to your comments, which we believe have greatly improved the manuscript.
Abstract:
Point1: Please specify the main overall purpose of this study.
Response 1: Thank you for this comment. We have clarified the main objective of the paper in the abstract. See page 1, lines 17-19.
Introduction:
Point 2: Lines 54-55: please include a few words about main assumptions of “One Health approach” on the basis of references cited.
Response 2: We appreciate this recommendation and have expanded this phrase to stress the importance of this perspective for the future health of the planet. See page 2, lines 60-66.
Point 3: Lines 56-57: please consider addition in this part of the text more precise description of potential non-clinical factors involved in prescription of ATB
Response 3: Thank you for your advice. We have expanded the text in this regard. Certain factors and attitudes, such as fear, complacency and the lack of appropriate legislation have been proven to influence the prescription of antibiotics in human medicine. Thus, it should be researched whether there are factors (independent of the clinical or scientific conditions which justify them) which may be influencing an inappropriate use of antibiotics by veterinarians. See page 2, lines 67-73.
Point 4: Lines 60-61: please consider addition into this part the main similarities and differences between factors influencing decisions of ATB prescription in humans and animals.
Response 4: Thank you for your comment. We understand that in spite of being extremely different sectors, the prescription of antibiotics may be influenced by factors which are common to veterinarians, doctors and pharmacists and by socio-demographic, economic, circumstantial or educational factors. Therefore, this study seemed relevant to us. However, we clarify in the paper that there are different factors and the prescription of antibiotics to animals may be justified by different reasons to the case of humans. See page 2, lines 71-80.
Results:
Point 5: Lines 183-185: please describe possible relation between ATB use and organization of farms. Are some data concerning use of ATB in farming systems such as agri-environment schemes or organic farms available? Did the authors observe any relation between types of practice (pets, horse industry, animal production) and evaluated indices?
Response 5: Thank you for your suggestion. We present one of the many examples which we have found of how an inappropriate organisation in farms facilitate the expansion of infectious diseases, thereby increasing the use of antibiotics. For example, the lack of an isolation circuit for sick animals, the lack of an appropriate udder cleaning circuit, the inadequate circulation of sewage, the availability of appropriate circuits for the entrance and exit of workers, etc. With regard to the type of practice, it is true that these factors stand out particularly in the livestock sector and not in the case of household pets. However, we have not found common and consistent indices in order to be able to compare these factors among different sectors and regions. We have added an example in this section and have commented on the difference between the two main sectors. See page 13, lines 194-199.
Point 6: Lines: 188-190: are there some specific factors for example involved in organization of veterinary clinics or farm specific?
Response 6: Thank you for your question. With regard to organisation, veterinarians working on farms frequently complained about the circuits, the hygiene programmes, the environmental biosecurity controls, etc. Among veterinarians working with household pets, the most common complaints with regard to organisation referred to a lack of professionals or an excess of work, both of which, in their opinion, could lead to errors in prescription. See page 13, lines 200-204.
Discussion
Point 8: Lines 270-275: due to the serious discrepancy in the results achieved, it may be worth clarifying whether the values depended on age were additionally related to the country or type of veterinary practice or other factors
Response 8: This is a very good suggestion. All of the studies (except one) in which the experience of the veterinarian was evaluated were contextualised in developed countries and in the livestock sector and the only study which evaluated this factor set in the context of pets did not reveal conclusive results. Therefore, we cannot make any additional analyses. However, we have decided to reflect this in the paragraph. See page 15, lines 289-291.
Point 9: Lines 412-425: Which part of results is discussed here? Please indicate
Response 9: Thank you. In this paragraph, we wanted to reflect on the cost and delay in obtaining the results of the antibiograms and the cultures. These two factors were mentioned in many of the studies as a limitation for a correct prescription. We have clarified in the text that these are the results that we are discussing. See page 18, lines 439-440.
Point 10: Lines 427-440: please take into account and describe the question how the formulation of antibiotic and the route of administration influenced the misuse (as in case of ototopical ointments composed of several antibiotics).
Response 10: We see your point and have included a phrase to make this relationship easier to understand. Thank you. See page 18, lines 455-457.

Round 2
Reviewer 1 Report
Thank you for taking most of my suggestions on board. I believe the paper has much improved, is better to understand, showing more clearly the limitation and taking a much more balanced 'One Health' approach. Congrats.
The only issue remaining is please update reference 10 (also on line 41-44: to the latest 2020 ESVAC report).